# Onion Polyphenols as Multi-Target-Directed Ligands in MASLD: A Preliminary Molecular Docking Study

**DOI:** 10.3390/nu16081226

**Published:** 2024-04-20

**Authors:** Maria Rosaria Paravati, Anna Caterina Procopio, Maja Milanović, Giuseppe Guido Maria Scarlata, Nataša Milošević, Maja Ružić, Nataša Milić, Ludovico Abenavoli

**Affiliations:** 1Department of Health Sciences, University “Magna Graecia”, 88100 Catanzaro, Italy; mrparavati@unicz.it (M.R.P.); procopioannacaterina@gmail.com (A.C.P.); giuseppeguidomaria.scarlata@unicz.it (G.G.M.S.); 2Department of Pharmacy, Faculty of Medicine Novi Sad, University of Novi Sad, Hajduk Veljkova 3, 21000 Novi Sad, Serbia; maja.milanovic@mf.uns.ac.rs (M.M.); natasa.milosevic@mf.uns.ac.rs (N.M.); natasa.milic@mf.uns.ac.rs (N.M.); 3Faculty of Medicine, University of Novi Sad, Clinic for Infectious Diseases, University Clinical Centre of Vojvodina, Hajduk Veljkova 1, 21000 Novi Sad, Serbia; maja.ruzic@mf.uns.ac.rs

**Keywords:** MASLD, liver, onion, nutraceutics, in silico research, molecular docking

## Abstract

A sedentary lifestyle associated with unregulated diets rich in high-calorie foods have contributed to the great prevalence of metabolic dysfunction-associated steatotic liver disease (MASLD) latterly, with up to 60% in the high-risk population and 25% in the general population. The absence of specific pharmacological strategies for this syndrome represents one of the major problems in the management of MASLD patients. Lifestyle interventions and adherence to a healthy diet are the main cornerstones of current therapies. The identification of nutraceuticals useful in the treatment of MASLD appears to be one of the most promising strategies for the development of new effective and safe treatments for this disease. The onion, one of the most widely studied foods in the field of nutraceuticals, serves as an inexhaustible reservoir of potent compounds with various beneficial effects. The following preliminary study analyzes, mediating in silico studies, the iteration of a library of typical onion compounds with 3-hydroxy-3-methylglutaryl-coenzyme A reductase, liver receptors X α and β, as well as peroxisome proliferator-activated receptors α and γ. In this study, for the first time promising smart molecules from the onion that could have a beneficial action in MASLD patients were identified.

## 1. Introduction

The term non-alcoholic fatty liver disease (NAFLD) was used in 1986 by Schaffner, who observed that the inflammatory state typical of non-alcoholic steatohepatitis (NASH) was not present in NAFLD [1]. NAFLD is identified as a syndrome characterized by fat accumulation in the liver parenchyma and encompasses a broad spectrum of pathologies including simple fatty liver, non-alcoholic steatohepatitis, cirrhosis, and hepatocellular carcinoma (HCC) through progressive fibrosis [2,3]. Recently, it has been proposed that the definition of fatty liver disease associated with metabolic dysfunction (MASLD) may represent a more comprehensive terminology than the acronym NAFLD since it overarches the relationship between hepatic fat deposition and metabolic dysfunction. Up to now, no specific drugs have been identified for the treatment of MASLD patients. The management of MASLD patients involves changes in lifestyle and the consumption of a healthy diet [4]. The Mediterranean diet is recognized for both the decrease in the cardiovascular diseases risk and for the reduction in the metabolic syndrome risk and, thus, it has been an efficient therapeutical approach in MASLD patients [5,6]. In this regard, the scientific community has made considerable efforts in identifying nutraceutical compounds useful in MASLD patients. Onion (*Allium cepa*) is believed to have originated in Central Asia, but it is farmed globally and eaten regardless of the cuisine and ethnic group. Cooked, fired, or raw, the onion is an irreplaceable ingredient of the Mediterranean diet. This perennial or biennial bulbous monocotyledon which belongs to the Liliaceae family is rich in various nutraceutical compounds, such as flavonoids, anthocyanins, organosulphur compounds, saponins, and phenols. Numerous favorable properties are attributed to the nutraceutical compounds in onions, including antioxidant, antitumour, anti-diabetic, and anti-inflammatory activities [7]. Among the different nutraceuticals, quercetin, kaempferol, myricetin, isorhamnetin, galangin, baicalein, and luteolin are recognized as promising antioxidative and anti-inflammatory agents. Nutraceuticals are “a food or part of a food that provides benefits health in addition to their nutritional content” [8]. For this reason, their potential medical use in gut–liver axis diseases were investigated. Quercetin is naturally occurring in a variety of foods such as apples, berries, brassica vegetables, onions, tea, tomatoes, and nuts. This flavonol has been identified for its ability to improve fibrosis development by reducing the production of Tumor Necrosis Factor (TNF), interleukin (IL)-6, IL-1β, and IL-8. Additionally, it enhances antioxidant mechanisms mediated by glutathione (GSH) and IL-10 while reducing lipid peroxidation in alcoholic liver disease (ALD) [9]. Kaempferol, which is found in tea, broccoli, apples, strawberries, and beans, prevents tumor development by deactivating PI3K/Akt/mTOR signaling, thus inhibiting migration, proliferation, and invasion. Furthermore, kaempferol inhibits cytochrome P450 2E1 (CYP2E1), protecting hepatocytes against ALD development, and attenuates fibrosis development by inhibiting SMAD2/3 through direct interaction with the ATP-binding pocket of activin receptor-like kinase 5 [10]. Myricetin, found in honey, vegetables, and wine, has a regressive effect on steatosis development in preclinical NASH models by promoting NRF2-mediated mitochondrial functionality, thereby increasing antioxidative enzyme activities and peroxisome proliferator-activated receptor (PPAR)-mediated fat decomposition [11]. Isorhamnetin is present in pears, onions, olive oil, and tomatoes. It alleviates steatosis by reducing FAS activity and fibrosis development via transforming growth factor -β-mediated hepatic stellate cell activation and proliferation, while decreasing the production of lipoperoxide compounds in serum and the liver [12]. Galangin is less abundant in nature, primarily found in galangal rhizomes and propolis. This polyphenol-mediated nuclear factor erythroid 2–related factor 2 (NRF2) activation attenuates oxidative damage, inflammation, and apoptosis during hepatoxicity, while inhibiting the proliferation of HCC cells through the combined activation of NRF2 and heme oxygenase-1 [13,14]. Baicalein, present in Walsh onion, effectively reduced hepatic fat accumulation both in vitro and in vivo by enhancing AMPK activation and suppressing sterol regulatory element binding transcription factor 1 (SREBP1) cleavage, consequently inhibiting the transcriptional activity of SREBP1 and the synthesis of hepatic fat. Additionally, baicalein exhibited total cholesterol lowering, increasing high-density lipoproteins (HDL), lowering low-density lipoproteins, as well as antioxidant and anti-inflammatory activities [15]. Finally, luteolin, present in a broad range of vegetables, fruits, and grains, alleviates and modulates obesity-associated NAFLD in mice models. Indeed, it can suppress the hepatic conversion of excess carbohydrates to triglycerides (TG) by inhibiting the LXR-SREBP-1c pathway [16]. These compounds belong to the class of flavonoids, specifically to the subclass of flavonols.

The main chemical structure of flavonoids involves two phenyl groups connected by a three-carbon atom bridge. This three-carbon atom bridge may be an open linear chain or it may be involved in a heterocyclic ring. Regardless of the structure it takes on, the three-carbon atom bridge exhibits a ketone function. In the specific case of flavonols, the three-carbon atom bridge forms a third heterocyclic ring, in which an unsaturation is also present. The three rings present in the chemical structure can then have various hydroxyl groups as substituents, which characterize the different flavonols [17]. Thus, the nutraceutical compounds from A. cepa may be observed as potential ligands for carefully selected target proteins that interfere with MASLD development and progression. Promising targets for the identification of new pharmacological strategies useful in MASLD include 3-hydroxy-3-methylglutaryl-coenzyme A reductase (HMGCR), PPARs, and liver X receptors (LXRs). HMGCR is an enzyme involved in the biosynthesis of cholesterol. HMGCR catalyzes the reduction reaction of (S)-HMG-CoA to (R)-mevalonate, using two molecules of NADPH [18]. Statins, as hypolipidemic drugs, are the main inhibitors of HMGCR. Currently, the usefulness of statins in MASLD patients has been demonstrated due to their ability to reduce liver enzyme concentrations and improve the liver histology of patients. Furthermore, it has been observed that statins reduce the risk of developing MASLD [19]. On contrary, the administration of statins is related to side effects such as the increase in liver enzymes as markers of inflammation. PPARs comprise three subtypes: PPARα, PPARδ, and PPARγ. The PPARα is involved in the processes of fatty acid absorption and oxidation and lipoprotein metabolism, and it is mainly expressed in the liver, the kidney, the heart, and the muscles. PPARδ is observed in most cells, while PPARγ is expressed in macrophages, the large intestine, and adipose tissue, where it is involved in adipocyte differentiation processes and lipid metabolism. Fibrates, by activating PPARα, promote the formation of HDL and reduce the concentration of TG in the blood and, thus, are the drugs of choice in dyslipidaemia [20]. Given the therapeutic effects of fibrates, the representatives of the new generation, saroglitizar and lanifibranor, are in clinical trials for MASLD and seem to meliorate NAFLD and NASH [21]. Thiazolidinediones (TZDs), PPARγ agonists, are mainly used in the treatment of type II diabetes due to their insulin-sensitizing action and ability to reduce plasma glucose levels. TZDs, and in particular pioglitazone, improve liver histology in NAFLD patients, although their application is limited due to the side effects. LXRs (α and β isoforms) are nuclear receptors involved in many metabolic processes, such as cholesterol homeostasis, liponeogenesis, and the inflammatory response. The LXRα isoform is expressed in tissues with high metabolic activity (liver, small intestine, and kidney), whereas LXRβ can be found ubiquitously [22]. Recent studies suggested that LXRs play a role as gatekeepers in NAFLD/NASH progression. Activation of LXRβ in stellate cells exerted anti-inflammatory and antifibrotic activities, preventing progression to fibrosis, while activation of LXRs in hepatocytes suppressed the transactivation of genes that promote NASH. LXRs are promising molecular targets for MASLD by virtue of the numerous cellular processes in which they are involved, but the complexity of their involvement in the disease requires further study. Hence, in this study, a library of naturally occurring onion compounds were used and molecularly docked on carefully selected molecular targets involved in the complex mechanism of MASLD onset and progression. The aim of the research was to select a small molecule/polyphenol that could interact with different targets involved in MASLD, based on a concomitant multiple-target approach.

## 2. Materials and Methods

The library of typical natural onion compounds used in the study was created by consulting FOODB and PubChem online databases [23,24] and the following compounds were selected: quercetin (2-(3,4-dihydroxyphenyl)-3,5,7-trihydroxychromen-4-one); kaempferol (3,5,7-trihydroxy-2-(4-hydroxyphenyl)chromen-4-one); galangin (3,5,7-trihydroxy-2-phenylchromen-4-one); baicalein (5,6,7-trihydroxy-2-phenylchromen-4-one); luteolin (2-(3,4-dihydroxyphenyl)-5,7-dihydroxychromen-4-one); myricetin (3,5,7-trihydroxy-2-(3,4,5-trihydroxyphenyl) chromen-4-one); isorhamnetin (3,5,7-trihydroxy-2-(4-hydroxy-3-methoxyphenyl)chromen-4-one); and (R)-2-(3,4-dihydroxybenzoyl)-2,4,6-trihydroxy-3(2H)-benzofuranone. In particular, the compound benzofuranone was found in “Rossa di Tropea” and “Ramata di Montoro” onions, two Italian onion varieties [25]. The 2D structures of the analyzed compounds were created using ChemDraw Professional 16.0 software (Figure 1) [26]. Optimization of the structures was carried out by applying the MMFF94 force field in Chem3D 16.0 software [27].

The X-ray crystallographic patterns of the selected targets were obtained from the Protein Data Bank (PDB): LXR-α (PDB: 1UHL); LXR-β (PBD: 1UPV); PPAR-α (PDB: 8HUK); PPAR-γ (PDB: 6D8K); and HMG-CoA reductase (PDB: 1HW8). The PDBs were optimized using the Wizard tool of the GOLD software v. 2022.3.0. GOLD allows flexible ligand and rigid receptor docking. The optimization enabled the addition of missing H-atoms and the removal of water molecules. The selected targets were re-docked using co-crystallized ligands. In particular, re-docking was performed with the antagonist mevastatin for HMGCR, and with the agonists T0901317, lanifibranor, and GW1929 for LXR-α/β and PPARα/γ, respectively. The GOLD Software’s Wizard tool enabled the identification of the binding pocket of the selected target [28]. The chemscore kinase and slow algorithm represent the parameters applied in our analysis. The library of selected compounds was molecularly docked, reporting the compounds with the best ChemPLP Fitness scores, and the interactions were analyzed with the academic version of Maestro, Schrödinger’s software v13.4 [29].

## 3. Results

The PDBs used in the study were selected considering the completeness of the amino acid structure of the crystallographic model and the activity of the co-crystallized ligand. Molecular modelling studies were performed in order to investigate and analyze the binding mode of the best docking poses of each compound, characterized by the best ChemPLP Fitness score (Table 1).

The protocol was validated by re-docking between the targets and their respective ligands. PDB 1HW8 represents the crystallographic model of HMGCR used in the study, in which the antagonist mevastatin constitutes the co-crystallized ligand. Box A in Figure 2 shows the re-docking results for mevastatin, highlighting the formation of two hydrogen bonds with Glu559 and Asp690 and three salt bridges with Arg590, Lys692, and Lys735 in the catalytic site A of HMGCR. The results are in agreement with data reported in the literature that attribute a crucial role to these interactions in HMGCR inactivation [30]. Boxes B, C, D, and E show the docking of benzofuranone, myricetin, quercetin, and luteolin, respectively. Analyses of the molecular docking of benzofuranone (ChemPLP Fitness score 58.726) showed the formation of five hydrogen bonds with Glu559, Lys691, Ser665, Asn755, Ala751 and hydrophobic interactions with Arg590 and His752, respectively. Molecular docking of myricetin (ChemPLP Fitness score 58.348) and quercetin (ChemPLP Fitness score 57.4561) showed the formation of hydrogen bonds with Ala751, Arg590, Gly560, Cys561, and Lys735 and two hydrophobic character interactions with Arg590 and Lys691. Luteolin (ChemPLP Fitness score 54.989) showed the formation of hydrogen bonds with Glu675, Lys691, Glu559, Asn755, Ala751, Lys735 and a hydrophobic interaction with Arg590.

PDB 1UHL represents the crystallographic model of LXRα used in the study, in which the synthetic agonist T0901317 constitutes the co-crystallized ligand. Box A in Figure 3 shows the re-docking results for T0901317, highlighting the formation of a hydrogen bond with the His421 of LXRα. This interaction is in agreement with the data reported in the literature, which attribute to this bond a crucial role in LXRα activation [31]. Boxes B, C, and D show the docking of myricetin (ChemPLP Fitness score 55.083), luteolin (ChemPLP Fitness score 53.643), and kaempferol (ChemPLP Fitness score 51.474), respectively. These compounds exhibit hydrogen bonding to H421, reproducing the interaction observed in T0901317; they also exhibit hydrogen bonding to Trp443. Other protein–ligand interactions can increase ligand stability. For example, Thr302 and Phe254 form two and one hydrogen bonds with myricetin, respectively. In contrast, luteolin and kaempferol each form a π-π interaction with Phe326 and Phe257, respectively.

The PDB 8HUK represents the crystallographic model of PPARα used in the study, in which the lanifibranor agonist constitutes the co-crystallized ligand. Box A in Figure 4 shows the re-docking results for lanifibranor. By superimposition between the crystallographic model and the pose obtained by re-docking, it was observed that the agonist retains the correct position (Root Mean Square Deviation (RMSD) value: 1.871Å). Boxes B and C show the docking of myricetin (PLP Fitness score 65.655) and galangin (PLP Fitness score 60.412), respectively. These compounds exhibit hydrogen bonding to Tyr464, reproducing the interaction reported in the literature, responsible for PPARα activation [32]. In addition, both compounds form hydrophobic interactions with His440 and Lys358. In addition, myricetin has hydrogen bonds with Ser280 and Phe351. This interaction network increases the stability of ligands at the binding site.

Molecular docking results for LXRβ and PPARγ were not reported in the study, as none of the compounds in the considered library reproduced the crucial interactions for receptor activation.

## 4. Discussion

MASLD is the most prevalent liver disease, reaching pandemic dimensions, and the leading cause of liver-related morbidity and mortality [33]. The lack of specific drug therapy for MASLD together with the inability to achieve and access the clinical endpoints due to the slow progression of the diseases is recognized as the main problem in MASLD management. The ideal therapeutical approach to MASLD should also target the metabolic risk factors that interfere with cardiovascular disease beside the mandatory inhibition of liver fibrosis progression [34]. Healthy lifestyle and a balanced diet are the most effective strategies to improve patients’ conditions. In our work, we identified five molecular targets involved in MASLD (HMGCR, LXRα/β, and PPARα/γ) that were subjected to molecular docking process, using a library of typical onion compounds. The docking poses obtained were analyzed by searching for protein–ligand interactions reported in the literature, which correlated with the activation of LXRα/β and PPARα/γ and the inhibition of HMGCR. This molecular docking study has made a pioneering step in analyzing a library of typical A. cepa compounds as potential LXRα/β and PPARα/γ activators and HMGCR inhibitors.

HMGCR is an enzyme involved in cholesterol biosynthesis and is responsible for the key reaction in the biosynthetic process. HMGCR catalyzes the conversion of HMG-CoA to mevalonate, which is the precursor of cholesterol and intermediate isoprenoids, such as farnesylpyrophosphate (FPP) and geranylgeranyl pyrophosphate (GGP). FPP and GGP act as lipophilic anchors on the cell membrane for GTPase proteins Ras and Rho, coordinating several cellular and molecular processes like cell survival, proliferation, and motility [35]. Reducing HMGCR activity can decrease the amount of cell-associated cholesterol, activating SREBP-2-mediated signaling pathways. Consequently, the upregulation of the low-density lipoprotein receptor (LDLR) on hepatocytes increases the removal of cholesterol-rich LDL particles from the bloodstream [36]. HMGCR downregulation can lead to cardioprotective effects by inhibiting Rho protein in vascular smooth muscle cells, thus countering hypertrophy [37]. There is a close correlation between alterations in free cholesterol and the onset or progression of NAFLD [38]. Most MASLD patients have a 64% risk of cardiovascular disease, and, thus, inhibiting HMGCR can be of particular interest for therapeutic purposes due to reduced free cholesterol levels in the blood, decreased risk for atherosclerotic plaques, and anti-hypertrophic effects [39]. HMGCR consists structurally of two portions: the N-terminal portion resides in the membrane of the endoplasmic reticulum, while the C-terminal part is responsible for catalytic activity and is immersed in the cytoplasm. These two sections are connected via a linker region [40]. Specifically, the catalytic part is a homo-tetramer consisting of four identical amino acid chains. Each monomer consists of three domains: the N-domain is the smallest and forms the linker region with the portion in the endoplasmic reticulum; the L-domain is the largest and forms the binding site for HMG-CoA; and the S-domain forms the binding site for NADPH, which is necessary for catalytic activity. The four monomers unite to form dimers with two catalytic sites each. Statins are the main inhibitors of HMGCR. These drugs target the enzymes’ active sites and form a dense network of interactions with residues Arg590, Ser684, Asp690, Lys691, Lys692, Lys735, Asp755, Glu559, and Asp767 [41,42]. All analyzed compounds expressed good affinity for the catalytic site of HMGCR by generating a network of interactions with the surrounding amino acid residues. In particular, best ChemPLP fitness scores and the highest number of interactions with the enzyme were observed for benzofuranone, myricetin, quercetin, and luteolin. These data suggested a good stability of the compounds in the active site. Myricetin, quercetin, luteolin, and benzofuranone reproduced the typical interactions of statins with Arg590 and Lys691 at the catalytic site of HMGCR. Thus, a promising inhibitory activity for benzofuranone, myricetin, quercetin, and luteolin against HMGCR can be assumed. Therefore, we can hypothesize that the analyzed polyphenols reduce cholesterol synthesis by inhibiting the four catalytic sites of HMGCR. Furthermore, in agreement with pathways reported in the literature, inhibition of HMGCR by polyphenols would increase SREB-2 levels and, consequently, cause an up-regulation of LDLRs. The final effect that we hypothesize from our results consists of reduced serum LDL cholesterol levels and a cardioprotective effect. Several studies reported that quercetin, myricetin, and luteolin reduced HMGCR gene expression [43]. On the contrary, quercetin exhibited cholesterolaemic activity by directly inhibiting HMGCR in an animal model study [44]. In particular, the latter study would seem to be in line with our hypotheses. LXRα is a nuclear receptor involved in the regulation of cholesterol and lipid metabolism, liponeogenesis, and the inflammatory response. The role of LXRs receptors in MASLD has always been controversial. Recent studies have revealed an anti-inflammatory effect of LXRs, which could help reduce NASH and subsequent fibrosis. LXRs seem to act directly and indirectly on the inflammatory response by inhibiting it, although the pathways have not yet been clarified. Several studies have reported suppression of pro-inflammatory genes such as cyclooxygenase-2 (COX-2) and inducible NOS following the administration of LXRs agonists. Moreover, activation of LXRs receptors inhibits the Toll-like receptor (TLR) pathway by inducing the cholesterol transporter ABCA1. LXRs also diminish Nuclear Factor-kB (NF-kB) activity. Finally, studies in animal models reported that activation of LXRs inhibited the phosphoinositide-3-kinase cascade, reducing TNF-α gene expression and liver damage [45,46]. The structure of LXRα is characterized by five domains: the ligand-binding domain (LBD); the N-terminal activation domain; the DNA-binding domain (DBD); the hinge region; and the C-terminal domain. Specifically, the LBD consists of 10 α-helices and one β-sheet. Activation of the LXRα receptor is determined by the stabilization of α-helix 12 in the active conformation to facilitate co-activator binding. Oxycholesterols, endogenous agonists of LXRα, interact with the LBD by forming two bonds with His421 and Trp443. In contrast, synthetic agonists, such as T0901317, activate the receptor through the formation of a hydrogen bond with His421 alone, resulting in the stabilization of His421-Trp443 stacking [47]. The desirable chemical structural affinity between the LBD of LXRα and myricetin, luteolin, and kaempferol was reported in this study. All three compounds reproduced the binding mode of endogenous agonists, forming hydrogen bonds with both His421 and Trp443. In addition, each ligand was stabilized in the LBD by the formation of additional interactions that promote the stability of the receptor–ligand complex. From our results, we could expect that luteolin, myricetin, and kaempferol, activating LXRα, might regulate lipid balance and reduce the inflammatory response mediated by COX-2, inducible NOS, TLR, NF-kB, and the phosphoinositide-3-kinase cascade involved in NASH. Luteolin was already reported to improve hypercholesterolaemia and glucose intolerance through an up-regulation of LXRα in obese mice [48]. However, to the best of our knowledge, there is no literature data on the direct myricetin, luteolin, and kaempferol activity on LXRα. In addition, no studies have been reported on the anti-inflammatory effects derived from LXRα-mediated activation of the polyphenols analyzed. PPARα is a nuclear receptor involved in lipid metabolism in the liver through various pathways. Firstly, PPARα coordinates fatty acid uptake by directly regulating the expression of fatty acid transporters. Secondly, PPARα controls the expression of enzymes involved in β-oxidation of fatty acids by regulating their degradation. In addition, PPARα induces ketogenesis by up-regulating mitochondrial HMG-CoA synthase. Finally, PPARα is involved in the regulation of lipogenesis by indirectly inducing LXRα [49,50,51]. Interestingly, PPARα activates opposite processes according to the body’s needs. PPARα induces de novo lipogenesis after food intake while in the fasting phases, absorption, β-oxidation, and ketogenesis are promoted, in order to ensure proper energy supply to tissues [52]. Treatment with PPARα agonists resulted in increased insulin sensitivity in the obese mice model, suggesting a hepatoprotective effect of PPARα [53]. In addition, PPARα appears to improve NASH mainly due to its anti-inflammatory action. Its activation prevents intrahepatic lipid accumulation and inflammation, reducing the number of activated macrophages and hepatic stellate cells (HSCs) and improving the histological pattern typical of NASH. Hepatoprotective effects induced by activation of PPARα lead to increased fibroblast growth factor 21 (FGF21) levels both in the serum and in the liver. FGF21 is a hepatochin that is secreted into the bloodstream by the liver. By binding to a specific receptor, FGF21 improves systemic insulin resistance and lipid turnover. Thus, PPARα has pivotal role in preventing the accumulation of lipids in the liver and the progression to NASH due to the synergistic effects of all activities responsible for control of the energetic homeostasis, together with its anti-hyphenation properties [45,54,55].The structure of PPARα is characterized by five domains: activation function 1 (AF1) domains A and B, the DNA binding domain (DBD), the ligand binding domain (LBD), and the hinge region. Activation of PPARα involves the formation of a network of hydrogen bonds between the ligand and Tyr464, Tyr314, and Ser280 residues. These interactions stabilize the α-helix 12, facilitating coactivator binding. Fibrates are well-known PPARα agonists that activate the receptor through interaction with residues Tyr464, Tyr314, and Ser280 [56]. Based on the results obtained in this study, myricetin and galangin could be promising PPARα agonists, as they mimic some of the key interactions for receptor activation since both compounds form hydrogen bonds with Tyr464. Considering the effects triggered by the activation of PPARα, the intake of galangin and myricetin could reduce fatty acid concentrations and prevent intrahepatic lipid accumulation. In addition, enhanced insulin sensitivity, anti-inflammatory effects mediated by inactivation of macrophages and HSCs, and histological improvement due to the beneficial effects on the liver tissue of FGF21 could be expected. Six of the compounds in the library (banzofuranone, myricetin, luteolin, quercetin, kaempferol, and galangin) have promising activity for at least one of the selected targets. In particular, myricetin and luteolin appeared to be the most promising compounds from a multi-target perspective.

## 5. Conclusions

Eight typical onion compounds were molecularly docked onto five targets involved in MASLD in order to identify promising small molecules. Benzofuranone, myricetin, luteolin, and quercetin exhibited the most promising antagonist activity for HMGCR; myricetin, luteolin, and kaempferol might have favorable agonist activity towards LXRα; and finally, myricetin and galangin seem to have good agonist activity towards PPARα. In addition, myricetin and luteolin are recognized as potential multi-target-directed ligands in MASLD. Finally, the present pilot study provides interesting insights into the development of innovative therapeutic strategies. In particular, the study identifies promising polyphenols that could potentially be used to make dietary supplements from a multi-target perspective. In addition, the selected molecules could represent crucial scaffolds for the synthesis of functionalized organic molecules. Further studies will be needed to validate these hypotheses.

## Figures and Tables

**Figure 1 nutrients-16-01226-f001:**
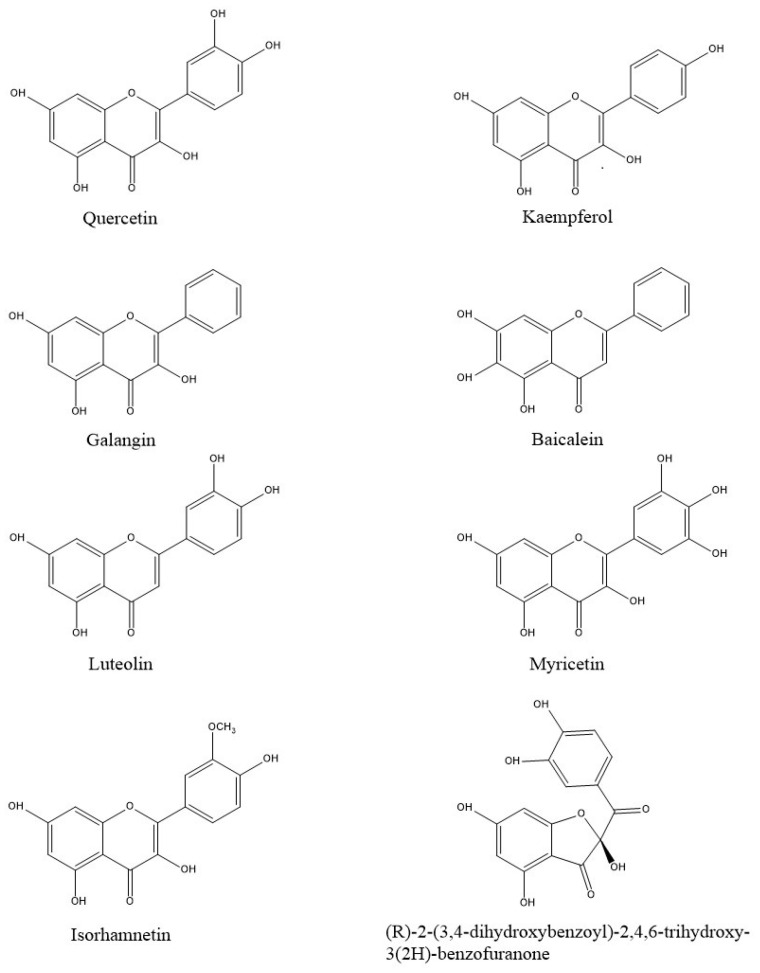
Two-dimensional structures of the ligands analyzed in this study.

**Figure 2 nutrients-16-01226-f002:**
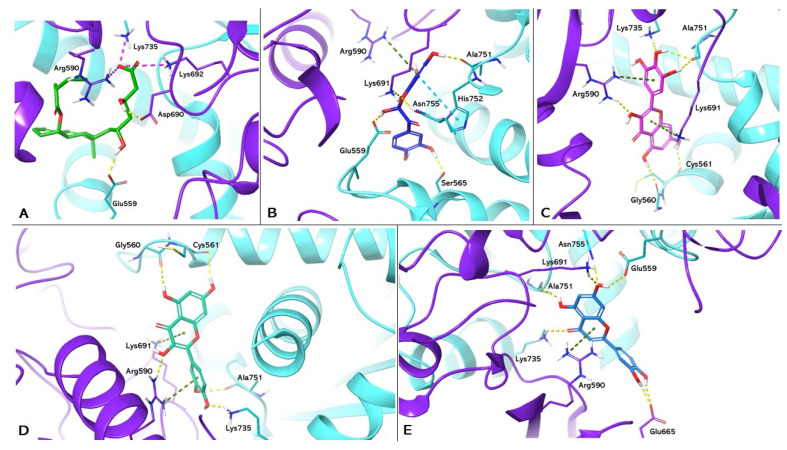
Molecular docking analysis. Three-dimensional representation of HGMCR complexed with mevastatin (**A**); benzofuranone (**B**); myricetin (**C**); quercetin (**D**); and luteolin (**E**) docked in the active site of HMGCR, respectively. HMGCR is shown as cartoon purple (Chain A) and cyan (Chain B). Mevastatin, benzofuranone, myricetin, quercetin, and luteolin are shown as light green, blue, magenta, dark green, and light blue carbon sticks, respectively. The amino acid residues involved in the interactions are shown as carbon sticks purple (Chain A) and cyan (Chain B), respectively. Hydrogen bonds, π-π, cation-π interactions, and salt bridges are shown as dashed lines, yellow, light blue, green, and magenta, respectively.

**Figure 3 nutrients-16-01226-f003:**
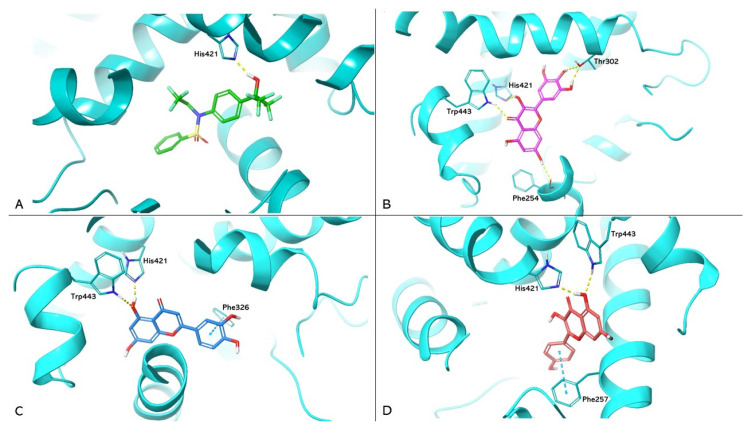
Molecular docking analysis. Three-dimensional representation of LXRα complexed with T0901317 (**A**); myricetin (**B**); luteolin (**C**); and kaempferol (**D**) docked in the LBD of LXRα, respectively. HMGCR is shown as cyan cartoon; T0901317, myricetin, luteolin, and kaempferol are represented as green, magenta, blue, and red carbon sticks, respectively. The amino acid residues involved in the interactions are shown as cyan carbon sticks. Hydrogen bonds and π-π interactions are shown as dashed lines, yellow and light blue, respectively.

**Figure 4 nutrients-16-01226-f004:**
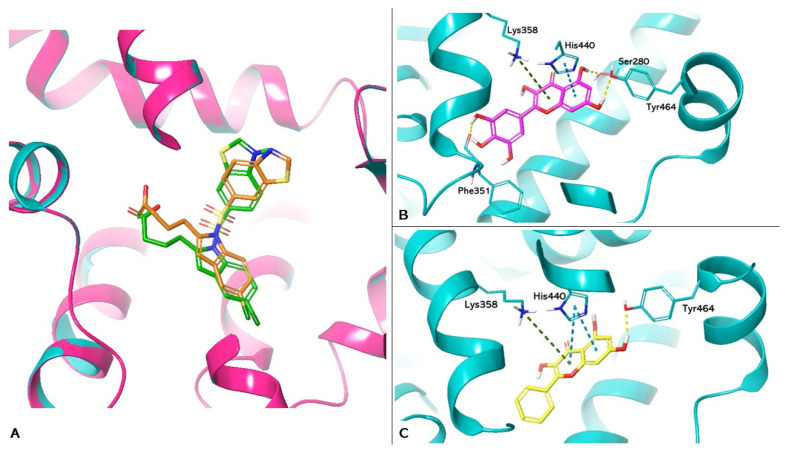
Molecular docking analysis. (**A**) Superimposition of the lanifibranor complexed with PPARα of the crystallographic model and the re-docking pose. PPARα is shown as magenta cartoons for the crystallographic model and cyan for the re-docking pose, respectively. Lanifibranor is shown as orange carbon sticks for the crystallographic model and green for the re-docking pose, respectively. (**B**,**C**) Three-dimensional representation of PPARα complexed with myricetin (magenta carbon sticks) and galangin (yellow carbon sticks) docked in the LBD of PPARα, respectively. PPARα is shown as cyan cartoon and the amino acid residues involved in the interactions are shown as cyan carbon sticks. Hydrogen bonds, π-π, and cation-π interactions are shown as dashed lines, yellow, light blue, and green, respectively.

**Table 1 nutrients-16-01226-t001:** ChemPLP Fitness score obtained from molecular docking of selected compounds on all targets, respectively.

Compound	LXR-α	LXR-β	PPAR-α	PPAR-γ	HMGCR
Quercetin	53.096	52.201	67.934	64.722	57.461
Kaempferol	51.474	52.585	58.782	65.393	51.729
Galangin	54.240	55.374	60.412	65.263	50.591
Baicalein	54.319	51.875	70.611	68.185	51.099
Luteolin	53.643	52.417	66.962	65.680	54.989
Myricetin	55.083	53.265	65.655	59.200	58.348
Isorhamnetin	54.087	50.041	54.434	62.746	50.958
Benzofuranone	52.215	52.927	58.102	69.113	58.726

## Data Availability

Data are contained within the article.

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
