# Peer review of "Onion Polyphenols as Multi-Target-Directed Ligands in MASLD: A Preliminary Molecular Docking Study"

_nutrients, 2024, doi:10.3390/nu16081226_

Round 1
Reviewer 1 Report
Comments and Suggestions for Authors
1) The analysis using molecular docking on each polyphenols that the author tested in this manuscript has been published in different journals.
For example, journals below
HMG – CoA reductase inhibition mediated hypocholesterolemic potential of myricetin and quercetin: in-silico and in-vivo studies
Insight into the mechanism of polyphenols on the activity of HMGR by molecular docking
So, retesting these molecules shown all together in the manuscript can not provide strong novelty.
2) people consume these polyphenols together as onion. So, the author need to test synergistic binding properties of each other (one to another, one to the other, one to two others etc) and then all together in the molecular docking system for each enzyme and each receptor.
This also should be designed with the amount of each compound in onion. For example, one compound is a very small amount in the onion compared to another compound. If this, this has a good binding property compared to the abundant compound, when people consume onion, this effect will not be strong and not be the same as the author predicted. So, this should be tested as well.
3) In MASLD, each enzyme and each receptor that the author tested in this study are inhibited or regulated by certain drugs (for example, statin the author already mentioned some of them, not all, but the author should test competitive molecular binding properties between each sample (single compound and combination etc as mentioned above) with each popular drugs as a positive control and also competitive binding with molecular docking system
4) the discussion should be seriously improved. The author did not talk about the detailed scientific regulation of each polyphenol in metabolisms for each enzyme and receptor that the author tested.
Author Response
1) The analysis using molecular docking on each polyphenols that the author tested in this manuscript has been published in different journals.
For example, journals below
HMG – CoA reductase inhibition mediated hypocholesterolemic potential of myricetin and quercetin: in-silico and in-vivo studies
Insight into the mechanism of polyphenols on the activity of HMGR by molecular docking
So, retesting these molecules shown all together in the manuscript can not provide strong novelty.
Thank you for your comment. Although few manuscripts have already been published, the conducted studies were focused mostly on potential of some polyphenols to inhibit HMG – CoA reductase. The novelty of this paper is a multi-target approach. The observed compounds, presented in onion, were tested on multiple receptors considering the complex mechanism of MASLD onset and progression. Studying multiple targets simultaneously, the aim of the research was to select small molecule/polyphenol that could interact with different targets involved in MASLD. Based on the presented results, myricetin and luteolin appeared to be the most promising compounds found in onion from a multitarget perspective. Moreover, as we reported in the paper, data for all polyphenols with all selected targets are not reported in the literature.
Introduction section was improved with the following lines: 39-43
2) people consume these polyphenols together as onion. So, the author need to test synergistic binding properties of each other (one to another, one to the other, one to two others etc) and then all together in the molecular docking system for each enzyme and each receptor.
This also should be designed with the amount of each compound in onion. For example, one compound is a very small amount in the onion compared to another compound. If this, this has a good binding property compared to the abundant compound, when people consume onion, this effect will not be strong and not be the same as the author predicted. So, this should be tested as well.
Thank you for your comment. The GOLD software used for molecular docking together with Maestro, Schrödinger's software do not allow a simulation of multiple ligands binding in cavity on the surface or inside of a protein that is suitable for ligand binding. Moreover, the molecular docking approach even assuming the use of other software, can only be used to predict the interaction between a single small molecule (in our case various polyphenols) and a protein at atomic level in order to elucidate the behavior of polyphenols in the binding pocket of target proteins [Meng XY, Zhang HX, Mezei M, Cui M. Molecular docking: a powerful approach for structure-based drug discovery. Curr Comput Aided Drug Des. 2011 Jun;7(2):146-57. doi: 10.2174/157340911795677602. PMID: 21534921; PMCID: PMC3151162.]. In order to significantly increase the docking efficiency, the binding sites of the co-crystallized ligands were used. In addition, the PDBs used in the study were selected considering the completeness of the amino acid structure of the crystallographic model and the activity of the crystallized ligand. Based on the basic principles of molecular docking, only prediction of the ligand conformation (position and orientation) within binding pocket, the explanation of the protein-ligand interactions and the assessment of the binding affinity expressed through a scoring function are possible in silico [Agu PC, Afiukwa CA, Orji OU, Ezeh EM, Ofoke IH, Ogbu CO, Ugwuja EI, Aja PM. Molecular docking as a tool for the discovery of molecular targets of nutraceuticals in diseases management. Sci Rep. 2023 Aug 17;13(1):13398. doi: 10.1038/s41598-023-40160-2. PMID: 37592012; PMCID: PMC10435576.]. Hence, the prediction of the agonist/antagonist activity of the single ligand, nor the synergistic effect of multiple ligands are not possible. These data can only be obtained by biological assays. Indeed, in this article based on the molecular docking approach, the information about the binding affinity of the different polyphenols in the binding pocket and the protein-ligand interactions were obtained and discussed. In our work, we hypothesized the agonist/antagonist activity of single ligands as a function of protein-ligand interactions known in the literature in activating or inhibiting the target. In this view, if the portion of the binding site is already occupied by a first compound, the presence of a second compound would provide no advantage. Our work is a preliminary study intended to serve as a pathfinder for further investigation.
Regarding the second point, based on the literature data onion can be considered as a rich dietary source of polyphenols (Metrani R, Singh J, Acharya P, K Jayaprakasha G, S Patil B. Comparative Metabolomics Profiling of Polyphenols, Nutrients and Antioxidant Activities of Two Red Onion (Allium cepa L.) Cultivars. Plants (Basel). 2020 Aug 21;9(9):1077. doi: 10.3390/plants9091077. PMID: 32825622; PMCID: PMC7569911.). Overall results are focused on total flavonoids, total phenolic content and total anthocyanins in different onions varieties. Moreover, the concentrations of individual polyphenols are still not available in the literature. In addition, the content of polyphenols differs between onion varieties and is strongly influenced by the cultivated methods and environment as main factors related to accumulation and concentration of polyphenols [Metrani R, Singh J, Acharya P, K Jayaprakasha G, S Patil B. Comparative Metabolomics Profiling of Polyphenols, Nutrients and Antioxidant Activities of Two Red Onion (Allium cepa L.) Cultivars. Plants (Basel). 2020 Aug 21;9(9):1077. doi: 10.3390/plants9091077. PMID: 32825622; PMCID: PMC7569911; Ren F, Reilly K, Gaffney M, Kerry JP, Hossain M, Rai DK. Evaluation of polyphenolic content and antioxidant activity in two onion varieties grown under organic and conventional production systems. J Sci Food Agric. 2017 Jul;97(9):2982-2990. doi: 10.1002/jsfa.8138. Epub 2017 Jan 12. PMID: 27859352.; Yang J, Meyers KJ, van der Heide J, Liu RH. Varietal differences in phenolic content and antioxidant and antiproliferative activities of onions. J Agric Food Chem. 2004 Nov 3;52(22):6787-93. doi: 10.1021/jf0307144. PMID: 15506817.]
In addition, the polyphenols analyzed are among the main constituents of onions, but they are also present in other foods. Hence, increased polyphenol concentrations could be met by consumption of polyphenol-rich foods or beverages [Scalbert, A., & Williamson, G. (2000). Dietary intake and bioavailability of polyphenols. The Journal of nutrition, 130(8), 2073S-2085S.] and/ or through the supplementation.
3) In MASLD, each enzyme and each receptor that the author tested in this study are inhibited or regulated by certain drugs (for example, statin the author already mentioned some of them, not all, but the author should test competitive molecular binding properties between each sample (single compound and combination etc as mentioned above) with each popular drugs as a positive control and also competitive binding with molecular docking system
Thank you for your comment. Molecular docking approach allows a calculation of a theoretical binding affinity between the studied ligand and the target binding pocket. We have already indicated the inability of in silico tools in determining the synergistic effects between two or more ligands inside the binding pocket. In addition, molecular docking does not allow the determination of a competitiveness between the popular drug and studied compounds. As was previously explained (replies to comment#2) only the scores values obtained through molecular docking of individual compounds towards targeted receptors could be compared and expressed as the chemical affinity for the receptor pocket. Based on the molecular docking methodology as a positive control reproduction of the same three-dimensional conformation of the drug observed in the crystallographic model of selected target should be performed. Hence, PDB 1HW8, used in this study, represents the crystallographic model of HMGCR in which the antagonist mevastatin constitutes the cocrystallised ligand. PDB 1UHL is the crystallographic model of LXRα, in which the synthetic agonist T0901317 constitutes the cocrystallised ligand while the PDB 8HUK represents the crystallographic model of PPARα, in which the lanifibranor agonist constitutes the cocrystallised ligand. We did our best to discuss the obtained interactions between studied polyphenols and targeted proteins and highlight the same mode of interactions with some drugs found in the literature.
4) the discussion should be seriously improved. The author did not talk about the detailed scientific regulation of each polyphenol in metabolisms for each enzyme and receptor that the author tested.
Thank you for your comment. Discussions have been implemented in the following lines: 282-297; 318-322; 324-325; 327-337; 351-354; 357-359; 360-381; 390-394.
Reviewer 2 Report
Comments and Suggestions for Authors
Fatty liver disease has been a serious health problem since viral hepatitis has been eradicated.
The authors propose that a series of chemical compounds of plant origin can help to improve this pathology. The article is well written. I believe that the authors could increase the receptivity of the article by opening a door to the therapeutic application of their studies. Also the presentation of the article should be improved.
This study mainly solves the problem that Some components of the onion may be useful for the treatment of fatty liver.
Liver degeneration is a disease that is increasing in prevalence. New drugs may be of interest in its treatment. The authors identify new compounds that may be useful in the treatment of the disease.
This study makes some contributions to New drugs that could be of interest in treatment of liver fatty disease.
Conclusions are consistent with the evidence and arguments presented. More experiments in humans must be done in a future. references are appropriate.
Comments on the Quality of English Language
Minor changes in English must be done.
Author Response
Fatty liver disease has been a serious health problem since viral hepatitis has been eradicated.
The authors propose that a series of chemical compounds of plant origin can help to improve this pathology. The article is well written. I believe that the authors could increase the receptivity of the article by opening a door to the therapeutic application of their studies. Also the presentation of the article should be improved.
This study mainly solves the problem that Some components of the onion may be useful for the treatment of fatty liver.
Liver degeneration is a disease that is increasing in prevalence. New drugs may be of interest in its treatment. The authors identify new compounds that may be useful in the treatment of the disease.
This study makes some contributions to New drugs that could be of interest in treatment of liver fatty disease.
Conclusions are consistent with the evidence and arguments presented. More experiments in humans must be done in a future. references are appropriate.
Thank you for your comment. We improved the introduction (lines 39-43) and added future therapeutic perspectives in the conclusions (lines 407-412).